# p53 Is Regulated in a Biphasic Manner in Hypoxic Human Papillomavirus Type 16 (HPV16)-Positive Cervical Cancer Cells

**DOI:** 10.3390/ijms21249533

**Published:** 2020-12-15

**Authors:** Linhan Zhuang, Regina Ly, Frank Rösl, Martina Niebler

**Affiliations:** Division of Viral Transformation Mechanisms, Research Program Infection, Inflammation & Cancer, German Cancer Research Center (DKFZ), Im Neuenheimer Feld 242, D-69120 Heidelberg, Germany; linhanzhuang@zju.edu.cn (L.Z.); r.ly@dkfz.de (R.L.)

**Keywords:** human papillomavirus, cervical cancer, hypoxia, p53, senescence, autophagy

## Abstract

Although the effect of hypoxia on p53 in human papillomavirus (HPV)-positive cancer cells has been studied for decades, the impact of p53 regulation on downstream targets and cellular adaptation processes during different periods under hypoxia remains elusive. Here, we show that, despite continuous repression of HPV16 *E6/E7* oncogenes, p53 did not instantly recover but instead showed a biphasic regulation marked by further depletion within 24 h followed by an increase at 72 h. Of note, during *E6/E7* oncogene suppression, lysosomal degradation antagonizes p53 reconstitution. Consequently, the transcription of p53 responsive genes associated with senescence (e.g., *PML* and *YPEL3*) cannot be upregulated. In contrast, downstream genes involved in autophagy (e.g., *DRAM1* and *BNIP3*) were activated, allowing the evasion of senescence under hypoxic conditions. Hence, dynamic regulation of p53 along with its downstream network of responsive genes favors cellular adaptation and enhances cell survival, although the expression of the viral *E6/E7*-oncogenes as drivers for proliferation remained inhibited under hypoxia.

## 1. Introduction

Cervical cancer represents the fourth most commonly diagnosed cancer in women worldwide [1], and mainly the high-risk human papillomavirus (HR-HPV) 16 and 18 are the most frequent types causing this disease [2,3]. Since 2006, broad health care programs are using three prophylactic HPV vaccines to provide protection against HR-HPV-related cancers [4,5]. However, incidence and mortality rates of cervical cancer remain high in many parts of the world where effective vaccinations and screening programs are missing [1,6,7,8]. The available HPV vaccines cannot treat advanced stages of HPV-associated diseases. For cervical cancer, treatment still consists of surgery combined with either pre- or post-operative adjuvant chemo and/or radiation therapy [9,10].

Such treatment options remain problematic owing partially to the tumors’ strong resistance to these kinds of therapeutic interventions [11,12]. Therapy-resistant progeny cells apparently derive from hypoxic areas frequently detected in cervical cancer tissues. Consistent with this notion is also that certain biomarkers of hypoxia (e.g., carbonic anhydrase-IX) negatively correlate with the prognostic outcome of the disease [13,14,15,16,17,18,19,20].

Hypoxic HPV-positive cancer cells can enter a dormant state, characterized by an evasion of cellular senescence [21] and resistance to cell death [22]. However, the underlying mechanisms are still not fully understood.

The p53 tumor suppressor functions as a transcription factor responding to a plethora of cellular stresses including hypoxia. p53 is not only important for DNA repair [23,24], cell cycle arrest [23,25], and apoptosis [23,26], but also for senescence [27], autophagy [28], and metabolism [29]. p53 activity depends on its intracellular localization and stability, which are both regulated by diverse post-translational modifications such as phosphorylation or ubiquitination, respectively, mediated by various enzymes [30].

In cells infected with HR-HPVs, the viral oncoprotein E6 promotes the binding of p53 to the ubiquitin ligase E6-AP, causing its proteasomal degradation, hereby inhibiting apoptosis and contributing to malignant transformation [31,32]. Repression of E6 in turn restores p53, resulting in the induction of p53-dependent senescence and/or apoptosis in HPV-positive cells [27,33].

As with most solid tumors, cervical cancers often display strongly reduced oxygen (O_2_) content compared to the atmospheric 21%, with a median O_2_ concentration of 1.18% [13,34]. Although the effect of hypoxia on HPV-positive cells has been already examined [21,22,35,36], the role of p53 in cellular adaptation, especially in the early phase under hypoxia, is still elusive.

In the present study, the dynamics of p53 in HPV16-positive cancer cells during hypoxia were examined. Notably, despite continuous repression of HPV16 *E6/E7* oncogenes, p53 did not recover but instead showed a biphasic regulation (rapid further reduction followed by a marked recovery after prolonged hypoxia). Such a dynamic along with the impact on corresponding downstream p53 responsive genes may contribute to enhance cellular adaptation to protect hypoxic *E6/E7*-repressed cancer cells from an irreversible commitment towards apoptosis or senescence. Predominantly, autophagy as a downstream response is critical for the evasion of senescence under hypoxic conditions. These findings reveal a novel niche of a molecular mechanism where pharmacological drugs could act to prevent cancer cell survival and recurrence in poorly oxygenated regions of HPV-positive cancers.

## 2. Results

### 2.1. p53 Is Regulated in a Biphasic Manner in Hypoxic E6/E7-Repressed HPV16-Positive Cervical Cancer Cells

To investigate the effect of hypoxia on p53 in an in vitro system, HPV16-positive cervical cancer cells (SiHa, CaSki) were cultured at 21% O_2_ (normoxia) or 1% O_2_ (hypoxia) for 24 h. Here, despite a downregulation of HPV16 *E6/E7* oncogene expression by up to 90% under hypoxia, p53 was not restored and even diminished to almost non-detectable levels. At the same time, hypoxia-inducible factor 1α (HIF-1α) was readily induced, which in turn promoted transcription of vascular endothelial growth factor (*VEGF*) (Figure 1A,B), a known downstream target gene required for neovascularization [37].

Next, time-course experiments were carried out in order to monitor HPV16 *E6/E7* mRNA levels and the amount of p53 protein under hypoxic conditions, ranging from 6 to 72 h. As depicted in Figure 1C (upper panel), viral transcription was downregulated by roughly 90% in SiHa and 80% in CaSki cells. Examining the levels of p53, a biphasic regulation under hypoxic conditions was noted—first, a rapid and strong decrease (by 24 h) but subsequently a recovery to even higher levels than under normoxic conditions at 72 h (Figure 1C, lower panel). The protein levels were comparable to those after siRNA-mediated knockdown of *E6/E7* and remained stable until the end of the experiment at 120 h of hypoxic incubation (Appendix A).

### 2.2. Lysosomal Degradation Is Required for the Initial Reduction of p53 in Hypoxic Cells

To decipher the mechanism underlying the biphasic regulation under hypoxia, p53 transcription was analyzed. However, as shown in Figure 2A and Appendix A, changes of the amount of *TP53* mRNA in either cell line did not account for the quantitative differences discerned on protein level. Despite an initial decrease by 60% within the first 24 h of hypoxia, mRNA levels then remained rather stable throughout prolonged incubation under hypoxia of up to 72 h, while p53 protein started to fully recover to normoxic levels (Figure 1C and Appendix A).

To examine whether the reduction of p53 protein levels under hypoxia was due to changes in its half-life, SiHa and CaSki cells were first incubated either under normoxic or hypoxic conditions for 4 h, followed by the addition of cycloheximide (CHX) to block *de novo* protein synthesis [38] for different periods of time. Notably, as shown in Figure 2 (panel B and C), short-term hypoxia reduced the half-life of p53 by > 45% in SiHa cells (from 8.6 to 4.6 min) and to a similar degree in CaSki cells (Appendix A), suggesting that the additional quantitative reduction is mediated by a post-translational degradation mechanism.

Next, we determined which cellular pathway was involved in the reduction of p53 under hypoxia, since E6/E7 oncoprotein suppression should allow p53 recovery. In cervical carcinoma cells, p53 is normally prone to proteasomal degradation by the E6 oncoprotein and E6-AP, a ubiquitin-dependent ligase [39,40]. In the absence of E6 or in HPV-negative cells, p53 is controlled by MDM2, another ubiquitin-dependent ligase that counteracts its quantitative level and function via an autoregulatory loop [41]. In any case, ubiquitinated p53 will ultimately be targeted for degradation via the 26S proteasome [42]. However, blocking the proteasome with the specific inhibitor MG132 for 8 and 24 h prior to harvesting could not fully restore p53 protein levels in hypoxic cells to quantities observed under normoxic conditions (Figure 2D and Appendix A).

Apart from the ubiquitin/proteasome system, autophagy-lysosome-dependent degradation is another important mechanism to control intracellular protein half-life [43]. The lysosomal inhibitors chloroquine (CQ) [44] and bafilomycin A1 (Baf A1) [45] were therefore used to assess the role of this pathway on the regulation of p53 under hypoxia. The adaptor protein p62/SQSTM1, a marker of autophagy, is a representative cargo protein sequestered into autophagosomes and degraded in the autolysosomes [46,47]. As shown in Figure 2E, both CQ and Baf A1 prevented the reduction of p62 under hypoxia, indicating that the degradation by autolysosomes was efficiently blocked under these conditions. In contrast, only CQ but not Baf A1 inhibited hypoxia-associated p53 degradation. Baf A1 blocked the fusion of autophagosomes and lysosomes yet only prevented the degradation of p62 but not p53. This suggests that, unlike p62, p53 was not sequestered into autophagosomes but was degraded via the lysosomal pathway. Although the treatment of CQ slightly increased *TP53* mRNA levels under both normoxia and hypoxia, the extent of induction was not significant (Appendix A). Furthermore, specific siRNA silencing of the autophagy-related genes *ATG12* and *p62/SQSTM1* did not restore p53 protein levels under hypoxia (Appendix A), additionally confirming that lysosomal degradation and not autophagy caused p53 degradation after *E6/E7* oncogene suppression during the first 24 h of hypoxia.

### 2.3. Hypoxic HPV16-Positive Cancer Cells Do Not Enter Senescence

Repression of *E6/E7* oncogenes in HPV-positive cells results in the rapid induction of cellular senescence [48], and p53 is a key regulator in this response [27,49,50]. In agreement with previous results [21], the RNAi-mediated reduction of *E6/E7* transcripts by 50% under normoxia (Figure 3A, left) led to a massive augmentation of p53 levels (Appendix A) as well as strong SA-β-Gal staining and the appearance of typical morphological signs of senescence, such as cellular enlargement and flattening [51] (Figure 3A, right). This is, however, in contrast to cells where *E6/E7* expression was reduced in response to hypoxia (Figure 1A), indicating that senescence was circumvented (Figure 3B and Appendix A).

Apart from apoptosis, activation of the p53 signaling pathway is a classical response leading to senescence when cellular stresses such as hypoxia or ionizing radiation occur. Senescence induction requires the expression of particular p53 target genes, such as *PML* and Yippee-like-3 (*YPEL3*) [50].

We therefore analyzed RNA levels of these target genes to assess whether they also display a biphasic regulation pattern, as seen for p53. In SiHa cells, the transcript levels of *PML* and *YPEL3* indeed decreased under hypoxia and showed only a recovery after prolonged oxygen depletion (Figure 3C). In hypoxic CaSki cells, *PML* mRNA levels also declined continuously, whereas the levels of *YPEL3* showed a response that mirrored p53 protein dynamics (Appendix A). These results demonstrate that p53-dependent senescence genes such as *PML* and *YPEL3* were not activated during the first 24 h of hypoxia.

### 2.4. p53-Responsive Genes Associated with Autophagy Are Induced in Hypoxic Cells

Since hypoxic HPV16-positive cancer cells appear to evade senescence, we hypothesized that the biphasic regulation of p53 under hypoxia has diverse effects on downstream pathways within the p53 network. Based on this assumption, additional p53-responsive genes known to be involved in different functional pathways and cellular outcomes were selected [52]. Notably, there was a subset of genes that showed an expression pattern consistent with the biphasic regulation observed in p53 protein levels. This group encompassed genes encoding apoptotic proteins (e.g., *BAX* and *PUMA*) [52] as well as genes involved in DNA repair (e.g., *XPC*) [53] (Figure 4A,B and Appendix A left). *GADD45A*, which is involved in cell cycle regulation, DNA nucleotide excision repair, and epigenetic modification [54,55], showed a slight increase in SiHa and a two-fold increase in CaSki cells at 24 h of hypoxia (Figure 4B right and Appendix A right). After prolonged hypoxia, however, *GADD45A* was strongly increased at 48 h in SiHa cells, whereas, in CaSki cells, the peak of mRNA levels was at 24 h and then decreased after further hypoxic incubation.

In another set, two p53-responsive genes promoting autophagy (*DRAM1* and *BNIP3*) [56,57] were induced by hypoxia. While the mRNA profile of *DRAM1* closely correlated with the levels of p53 with an initial decrease followed by a strong increase at 48 h of hypoxia, *BNIP3* already displayed a ten-fold upregulation at 24 h of hypoxia in SiHa cells. Furthermore, the mRNA levels of *TIGAR*, a gene related to the repression of autophagy [58], decreased at the same time as p53 protein levels in both SiHa and CaSki cells (Figure 4C and Appendix A). This indicates that autophagy is induced by hypoxia in HPV16-positive cancer cells in the absence of p53.

### 2.5. The Evasion of Hypoxic HPV16-Positive Cancer Cells from Senescence Can Be Attributed to the Induction of Autophagy

Autophagy is essential to suppress cellular stress and to induce a dormant state in cells by preventing senescence [59,60]. Hypoxia-induced autophagy is further linked to hypoxia tolerance and tumor cell survival [61]. These considerations raise the possibility that autophagy is involved in the evasion of hypoxic cancer cells from senescence. In fact, as depicted in Figure 5A, hypoxia induced autophagy in HPV16-positive cancer cells, which showed a transient increase of LC3-II and a reduction of p62, two markers of autophagy. In order to further assess the role of p53 in the context of autophagy in our experimental system, we ectopically expressed wild-type p53 in cells prior to hypoxic incubation. As shown in Figure 5B, autophagy was readily induced in untransfected control cells, since p62 was decreased. In contrast, in cells where p53 was reconstituted, p62 remained stable. This indicates that autophagy was inhibited by ectopic p53 expression.

To examine whether inhibition of autophagy induces senescence, HPV16-positive cancer cells were incubated with the autophagy inhibitor chloroquine (CQ). Since CQ is known to inhibit the activity of beta-galactosidase [62], we were not able to detect strong SA-β-Gal staining in this assay. We did, however, observe the characteristic morphological signs of senescence (e.g., cell enlargement and flattening) in the presence of CQ as well as the apparent inability to grow when switched back to normoxic conditions (Figure 5C and Appendix A), which strongly suggests a senescent state. Unlike in SiHa cells, where CQ had no apparent growth inhibitory effect under normoxia (Figure 5C), CQ showed strong inhibitory effects on CaSki cells already under normoxia that was further pronounced under hypoxic conditions (Appendix A). This indicates a different dependence of these cell lines on autophagy or lysosomal degradation of p53 [63,64].

In summary and based on these findings, biphasic p53 regulation under hypoxia along with the respective downstream genes represents a survival and protective strategy of HPV16-positive cervical cancer cells where autophagy is required at prolonged hypoxia to circumvent apoptosis and senescence (Figure 6).

## 3. Discussion

A microenvironment with reduced or low oxygen availability affects all types of cells on their way to malignancy [65]. Even in transformed cells, hypoxia selects for phenotypes that allow the survival of progenies [66]. Genome sequencing data recently demonstrated divergent mutational processes of normoxic versus hypoxic tumors [67]. In any case, the key protein that is always a central decisive hub within the cellular network is p53, therefore justifying its role as “guardian of the cell” [68].

In contrast to most human tumors, HPV-positive premalignant and malignant cells still harbor wild type p53 whose availability, however, is diminished by HPV E6 oncoprotein-mediated proteasomal degradation [31,32]. Hypoxic HPV-positive cancer cells can enter a dormant state, characterized by evasion of cellular senescence [21] and resistance to cell death [22]. However, the role of p53 at different time intervals during and after hypoxia and its function in the decision determining which cellular phenotype ultimately manifests and accumulate still remains elusive.

Based on this, it was therefore intriguing to see that p53 showed a biphasic regulation in HPV16-positive cancer cells. Although *E6/E7* oncogene expression was reduced early after hypoxic incubation, p53 did not instantly recover within 24 h, as was the case when viral gene expression was impaired after specific siRNA delivery (Appendix A). Instead, in spite of stably reduced *TP53* mRNA levels after 24 h of hypoxia, p53 protein re-appeared delayed upon prolonged oxygen deprivation (starting at 48 h) (Figure 1 and Appendix A), raising the question about the selective benefit for the cells where such a mechanism occurred.

Considering multi-step carcinogenesis as an evolutionary process, there are several rounds of selection that certainly strongly affect all aspects of cellular homeostasis (transcriptome, proteome, and epigenetic changes, respectively) of both premalignant and fully transformed cells simply to allow adaptation to different microenvironmental constraints [65]. Hence, hypoxic changes can select for progeny cells that acquire, for instance, a higher resistance toward radiation therapy and antineoplastic drugs [69].

Indeed, unlike in response to *E6/E7* depletion under normoxia, hypoxic cells in which the levels of oncogene mRNA were similarly reduced did not enter senescence (Figure 3A,B and Appendix A), since p53 and crucial p53-responsive genes remained reduced (Figure 3C and Appendix A). Hence, speaking in evolutionary terms, biphasic regulation of p53 allows cell survival by preventing apoptosis or senescence.

Mechanistically, early hypoxia causes a decrease in p53 half-life that could not be attributed to disproportionate proteasomal degradation or autophagy, since specific inhibition of both pathways was insufficient to reconstitute p53 (Figure 2 and Appendix A). Only treatment with the lysosome inhibitor chloroquine (CQ) fully recovered p53 levels. Notably, hypoxia can induce a time-dependent redistribution of lysosomes within the cells as well as the induction of lysosomal proteins [70,71]. This might result in a shift in proteins subjected to lysosomal degradation.

Although p53 was not degraded by autophagy (Figure 2E and Appendix A), its depletion coincided with the induction of autophagy-related genes (Figure 4C and Appendix A). Moreover, the fact that treatment with CQ or ectopic expression of p53 inhibited autophagy (Figure 2E and Figure 5B) and led to an irreversible growth arrest of hypoxic cells (Figure 5C and Appendix A) further substantiates the key role of p53 in these processes. The loss of p53 in chronically starved cancer cells can impair autophagic flux and ultimately result in apoptosis [72,73]. A proper oscillation of autophagic factors is therefore critical for cell survival and might additionally facilitate the loss of apoptosis-related gene expression. Hence, continued quantitative reduction of p53 after prolonged hypoxia might lead to an excessive autophagic fluctuation [74]. Conversely, restoration of p53 after prolonged hypoxia could be another strategy of HPV16-positive cancer cells to enhance cell survival by maintaining advantageous autophagic homeostasis and adjusting the rate of autophagy to a changing microenvironment under prolonged hypoxia.

Previous studies already showed that hypoxia readily induces autophagy in cancer cells, but not all cancers are susceptible to the inhibition of this process, making a generalized cancer treatment strategy impossible [75]. We also observed certain differences between CaSki and SiHa cells with respect to the onset of p53 depletion and its recurrence (Figure 1C), p53-responsive gene profiles (Figure 4 and Appendix A), as well as the cells’ susceptibility to CQ under only hypoxic (SiHa, Figure 5C) or hypoxic and normoxic conditions (CaSki, Appendix A). This might be due to differences in the HPV16 copy number and integration sites [76] or simply—as explained by the current discourse of personalized medicine—by the individual genetic background of the cancer patients the cells derived from [77].

Based on our findings, we propose the following model for the hypoxia-induced regulation of p53 and its consequences for the cell’s fate (Figure 6). Under normoxic conditions, the siRNA-mediated knock-down of *E6/E7* caused a rapid and strong recovery of p53, resulting in the induction of apoptosis or senescence in HPV16-positive cells. Within the first 24 h of hypoxia, however, *E6/E7* expression was repressed, p53 was subjected to lysosomal degradation, and p53-responsive genes associated with senescence and apoptosis were down-regulated, while autophagy-related factors were induced in a p53-dependent manner. Treatment with chloroquine overcame the cells’ resistance to senescence and caused an irreversible growth arrest in hypoxic cells. After prolonged hypoxia, p53 protein levels fully recovered similar to oncogene-depleted levels under normoxia by a thus far unidentified mechanism that did not cause cell death. Nonetheless, targeted perturbation of the p53/autophagy axis may pave the way for novel pharmacological strategies to prevent cancer recurrence in poorly oxygenated regions of HPV-positive cancers.

## 4. Materials and Methods

### 4.1. Cell Culture

HPV16-positive cervical carcinoma cell lines CaSki and SiHa were grown in Dulbecco’s Modified Eagle’s Medium (Sigma-Aldrich, Taufkirchen, Germany) containing 10% fetal calf serum (Thermo Fisher Scientific, Dreieich, Germany). The authenticity of the cells was confirmed by an in-house authentication service platform. PCR was applied to exclude possible contaminations with *Mycoplasma spec*. Cells were cultured under normoxia (21% O_2_, 5% CO_2_) or hypoxia (1% O_2_, 5% CO_2_).

### 4.2. Treatment of Cells with Chemical Compounds

Cells were treated with the following chemicals at 21% O_2_ or at 1% O_2_: bafilomycin A1 (Thermo Fisher Scientific, Dreieich, Germany); chloroquine and cycloheximide (Sigma-Aldrich, Taufkirchen, Germany); and MG132 (Merck, Darmstadt, Germany). Bafilomycin A1, cycloheximide, and MG132 were dissolved in DMSO, chloroquine was dissolved in H_2_O, and stock solutions were stored at −20 °C. As control, cells were incubated with the same volume of the respective solvent. The compounds’ final concentrations are mentioned in the figure legends.

### 4.3. siRNA Transfection

For siRNA-mediated knockdown of genes, cells were transfected using Lipofectamine 2000 Transfection Reagent (Thermo Fisher Scientific, Dreieich, Germany) according to the manufacturer’s instructions. The siRNA oligonucleotides were purchased from Thermo Fisher Scientific: Silencer^®^ Select Pre-designed siRNA *Atg12* (5′-GCAGCUUCCUACUUCAAUUTT-3′) and Silencer^®^ Select Pre-designed siRNA *SQSTM1/p62* (5′-CUUCCGAAUCUACAUUAAATT-3′). A siRNA oligonucleotide pool was used to block HPV16 *E6/E7* expression as described previously [21]; scrambled control siRNA (Silencer^®^ Negative Control #1 siRNA, Cat#: AM4611) was purchased from Thermo Fisher Scientific, Dreieich, Germany. At 48 h post transfection, the cells were cultured for 0–24 h at 21% O_2_ or at 1% O_2_. After respective incubation periods, total protein or RNA was extracted as described below.

### 4.4. Ectopic Expression of p53

The 1.0 × 10^6^ SiHa or CaSki cells were seeded in 6 cm cell culture dishes and incubated overnight. Cells were transfected using Turbofect in vitro Transfection Reagent (Thermo Fisher Scientific, Dreieich, Germany). Then, 1 µg of the respective expression plasmids (pPK-CMV-E3 empty vector or pPK-p53 wild type) were diluted in 300 μL of Opti-MEM (Thermo Fisher Scientific, Dreieich, Germany) and 4 μL of Turbofect mixed by vortexing. After incubation for 15 min at room temperature, the mix was added to the culture dishes. At 48 h post transfection, the cells were cultured for 0–24 h at 21% O_2_ or at 1% O_2_. Total protein was extracted as described below.

### 4.5. Protein Extraction and Western Blotting

Cells were washed once with 1x phosphate buffered saline (PBS). CSKI buffer (10 mM piperazine-N,N′-bis(2-ethanesulfonic acid) (PIPES), 100 mM NaCl, 1 mM Ethylenediaminetetraacetic acid (EDTA), 300 mM Sucrose, 1 mM MgCl_2_, and 0.5% Triton^®^ X-100, pH 6.8) freshly supplemented with 1× complete protease inhibitor cocktail (Roche, Mannheim, Germany), 1 mM dithiothreitol (DTT), 1 µM MG132, 1 mM NaF, and 0.2 mM Na_3_VO_4_ was added to the cells, which were scraped off and collected. The suspensions were incubated on ice for 30 min and mixed every 10 min. The lysates were centrifuged for 30 min at 13,000 rpm and 4 °C. Supernatants were quantified using the Bradford Assay Reagent (Bio-Rad Laboratories GmbH, Feldkirchen, Germany). In total, 40–80 μg of denatured proteins were used for western blotting. After transfer, filters were incubated with the following primary antibodies: anti-p53 (sc-126, Santa Cruz Biotechnology, Heidelberg, Germany); anti-Atg12 (4180, Cell Signaling Technology, Frankfurt am Main, Germany); anti-SQSTM1/p62 (ab56416, abcam, Berlin, Germany); anti-GAPDH (sc-25778, Santa Cruz Biotechnology, Heidelberg, Germany); anti-HIF-1α (610959, BD Biosciences, Heidelberg, Germany) [21]; and anti-LC3A (NB100-2331, Novus Biologicals, Wiesbaden, Germany). The following HRP-conjugated secondary antibodies were used: goat-anti mouse IgG (H + L) and goat-anti rabbit IgG (H + L) (115-035-003, Jackson ImmunoResearch, Camebridgeshire, United Kingdom). Immunoblots shown in this manuscript depict representative images of three independent experiments, respectively.

### 4.6. RNA Extraction and Reverse Transcription

RNA was extracted using the RNeasy Mini Kit (Qiagen, Hilden, Germany) according to the manufacturer’s instructions. The DNA digestion was performed using the RNase-Free DNase Set (Qiagen, Hilden, Germany). RNA concentrations were determined photometrically, and 1 µg of RNA was reverse transcribed into cDNA using RevertAid Reverse transcriptase (Thermo Fisher Scientific, Dreieich, Germany) and random primers (Roche, Mannheim, Germany) according to the manufacturer’s protocol. A 1:5 dilution of the generated cDNA was used for PCR analyses.

### 4.7. Quantitative Real-Time PCR Analyses

Quantitative real-time PCR (qPCR) was performed with the CFX96 Real-Time PCR Detection System (Bio-Rad Laboratories GmbH, Feldkirchen, Germany) using iTaq Universal SYBR Green Supermix (Bio-Rad Laboratories GmbH, Feldkirchen, Germany) according to the manufacturer’s instructions. The transcript levels of *18S* rRNA were assessed to allow normalization of samples. A complete list of primer sequences can be found in Appendix A.

### 4.8. Cycloheximide (CHX) Treatment under Normoxic and Hypoxic Conditions

Cells were incubated either at 21% O_2_ or at 1% O_2_ for 4 h, treated with cycloheximide (CHX, 10 µg/mL), and harvested after 0, 10, 20, and 30 min for western blotting as described above.

### 4.9. Senescence Assays

Cells were fixed and stained for SA-β-Gal activity as described [21].

### 4.10. Statistical Analysis

Statistical significance was determined by the two-tailed Student’s *t* test. *p*-values of ≤ 0.05 (*), ≤0.01 (**), or ≤0.001 (***) were considered statistically significant.

## Figures and Tables

**Figure 1 ijms-21-09533-f001:**
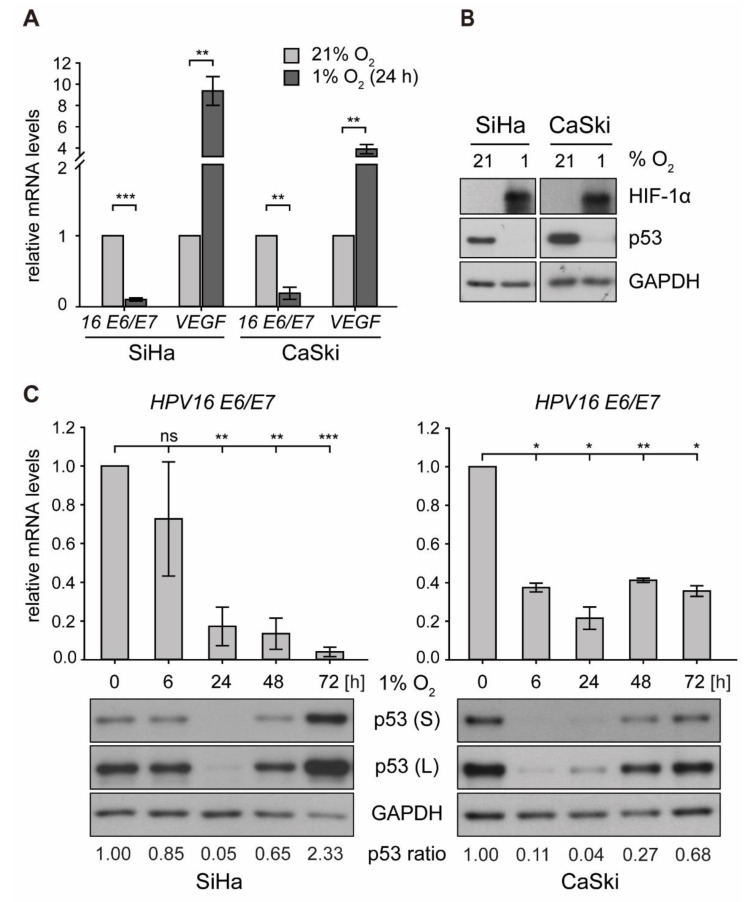
*E6/E7* oncogene transcription and p53 protein expression in human papillomavirus (HPV)16-positive cervical cancer cells under normoxia and hypoxia. Cells were cultured for the indicated times at 21% O_2_ or at 1% O_2_. Total RNA and total protein were extracted. (**A**) qPCR analyses of HPV16 *E6/E7* and vascular endothelial growth factor (*VEGF*) transcripts. The *18S* rRNA served as internal control for normalization. Error bars indicate the SD of three independent experiments. Corresponding transcript levels under normoxia were arbitrarily set to 1 and used as control for the calculation of significant differences (two-tailed Student’s *t*-test was employed, ** *p* < 0.01, *** *p* < 0.001). (**B**) Western blot analyses of cells after 24 h under normoxia/hypoxia. The filter was incubated with antibodies against p53 and hypoxia-inducible factor 1α (HIF-1α). Glyceraldehyde 3-phosphate dehydrogenase (GAPDH) served as a loading and transfer control. (**C**) Time course of hypoxic incubation. Upper panels: qPCR analyses of HPV16 *E6/E7* transcripts. The *18S* rRNA served as internal control for normalization. Error bars indicate the SD of three independent experiments. The 0 h value of *E6/E7* expression was set as a control for the calculation of significant differences (two-tailed Student’s *t*-test was employed, * *p* < 0.05, ** *p* < 0.01, *** *p* < 0.001 and ns, statistically not significant). Lower panels: Western blot analyses of p53 levels. GAPDH served as a loading control (S, short exposure; L, long exposure). The incubation time is indicated. The p53 ratio refers to normalized quantifications of p53 band intensities relative to GAPDH from three independent experiments (see Appendix A).

**Figure 2 ijms-21-09533-f002:**
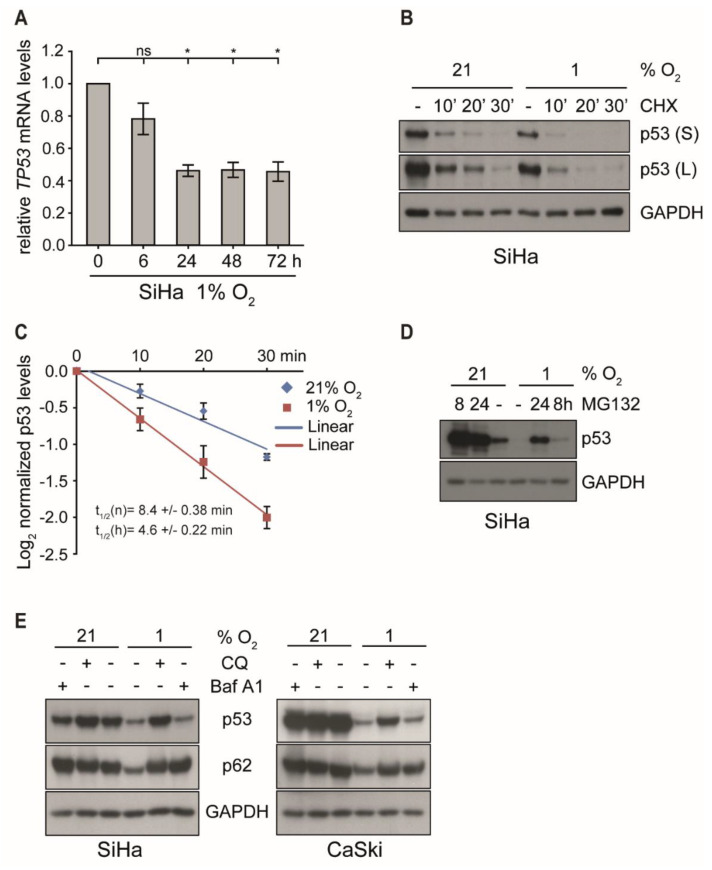
p53 transcription and protein stability under normoxic and hypoxic conditions in the presence or the absence of proteasomal and lysosomal inhibitors. (**A**) qPCR analysis of *TP53* transcripts. SiHa cells were cultured for the indicated times at 1% O_2_ prior to RNA extraction. The *18S* rRNA served as internal control. Error bars indicate the SD of three independent experiments. The 0 h value of *TP53* expression was set as a control for the calculation of significant differences (two-tailed Student’s *t*-test was employed, * *p* < 0.05 and ns, statistically not significant). (**B**) p53 levels in SiHa cells incubated at 21% O_2_ or at 1% O_2_ for 4 h and subsequently treated with/without cycloheximide (CHX, 10 µg/mL) for up to 30 min. Western blot analyses of p53. GAPDH served as a loading control (S, short exposure; L, long exposure). (**C**) The grayscale of p53 shown in (**B**) was quantified using the ImageJ software. The relative grayscale values of p53 normalized to the respective (21% O_2_ or 1% O_2_) 0 time point were plotted on an x/y diagram with log (2)-transformed *x*-axis to visualize the protein half-life. The median protein half-lives (t_1/2_) of normoxic (n) and hypoxic (h) samples are given. Error bars are SD, and *n* = 3 independent experiments. (**D**) p53 western blot analyses. SiHa cells were incubated at 21% O_2_ or at 1% O_2_ for 0 h or 16 h prior to the treatment with 10 μM MG132. Total protein was extracted after 24 h. GAPDH served as a loading control. (**E**) SiHa and CaSki cells were cultivated for 24 h under normoxia (21% O_2_) or hypoxia (1% O_2_) in either the absence (−) or the presence (+) of 50 μM chloroquine (CQ) or 0.5 μM Bafilomycin A1 (Baf A1). Western blot analyses of p53 and p62 were performed. GAPDH served again as a loading control.

**Figure 3 ijms-21-09533-f003:**
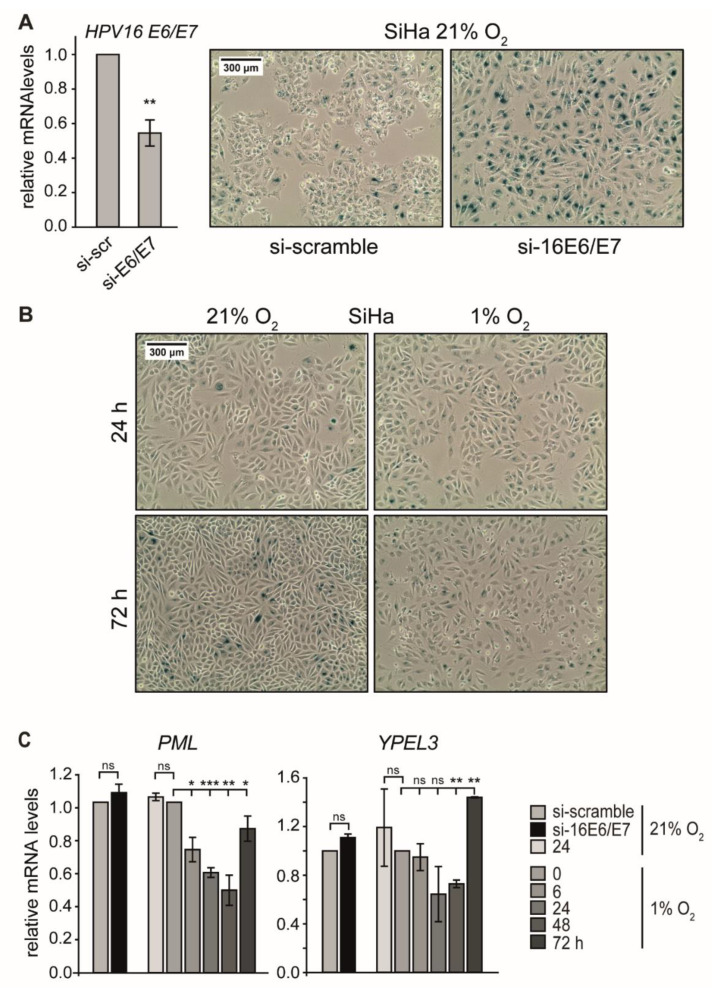
Effect of E6/E7 repression on senescence induction under normoxia and hypoxia. (**A**) SiHa cells were cultured at 21% O_2,_ and endogenous HPV16 *E6/E7* expression was silenced by RNAi. (A, left) qPCR analysis of HPV16 *E6/E7* mRNA levels following transfection with scrambled siRNA (si-scr) or siRNA directed against HPV16 *E6/E7* (si-E6/E7). The *18S* rRNA served as internal control. Error bars indicate the SD of three independent experiments. The scr value of HPV16 *E6/E7* expression was set as a control for the calculation of significant differences (two-tailed Student’s *t*-test was employed ** *p* < 0.01). (A, right) Cells were stained for the senescence marker SA-β-Gal expression at 72 h after siRNA transfection. Scale bar: 300 μm. (**B**) SiHa cells were cultured under normoxia (21% O_2_) or hypoxia (1% O_2_). After 24 h or 72 h, respectively, cells were stained for the expression of SA-β-Gal. Scale bar: 300 μm. (**C**) qPCR analysis of senescence-associated genes in SiHa cells. HPV16 *E6/E7* expression was silenced by RNAi under normoxia (si-16E6/E7), and total RNA was extracted 48 h after transfection. In parallel, cells were cultured for the indicated periods under normoxia (21% O_2_) or hypoxia (1% O_2_), and total RNA was extracted after different time intervals as indicated. The *18S* rRNA served as internal control. Error bars indicate the SD of three independent experiments. The si-scramble group or the 0 h hypoxia value of gene expression was arbitrarily set to 1.0 in order to calculate significant differences (two-tailed Student’s *t*-test was employed *** *p* < 0.001; ** *p* < 0.01, * *p* < 0.05 and ns, no significant difference).

**Figure 4 ijms-21-09533-f004:**
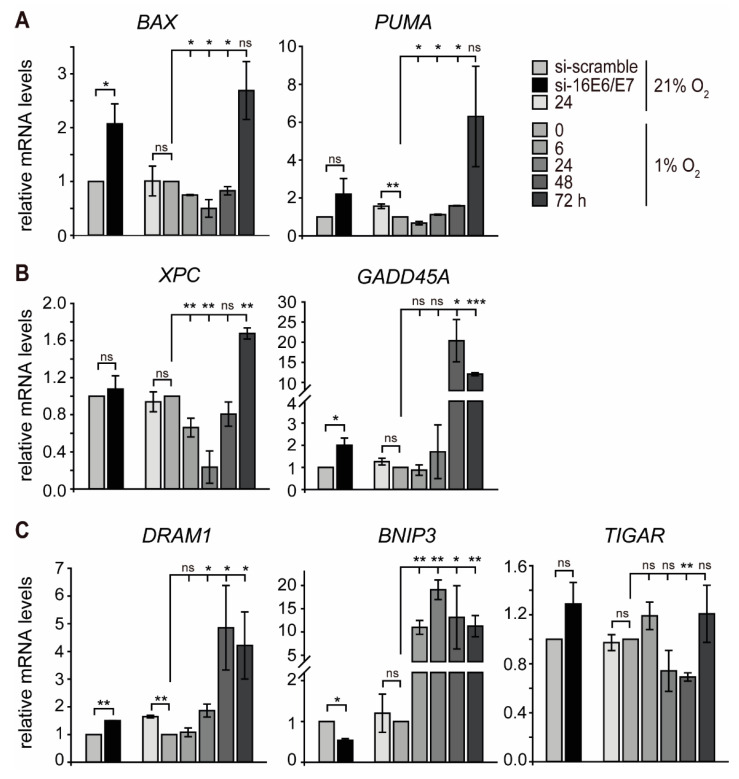
Transcript levels of p53-responsive genes in normoxic and hypoxic SiHa cells. HPV16 *E6/E7* expression was silenced by RNAi under normoxia (si-16E6E7), and total RNA was extracted 48 h after transfection. In parallel, cells were cultured under normoxia (21% O_2_) or hypoxia (1% O_2_), and total RNA was extracted as indicated. qPCR analyses of *BAX* and *PUMA* (panel **A**); *XPC* and *GADD45A* (panel **B**) and *DRAM1*, *BNIP3*, and *TIGAR* (panel **C**) transcripts. The *18S* rRNA served as internal control for normalization. Error bars indicate the SD of three independent experiments. The si-scramble group or 0 h hypoxia value of gene expression was set to 1.0 as a control to calculate significant differences (two-tailed Student’s *t*-test was employed, *** *p* < 0.001; ** *p* < 0.01, * *p* < 0.05 and ns, no significant difference).

**Figure 5 ijms-21-09533-f005:**
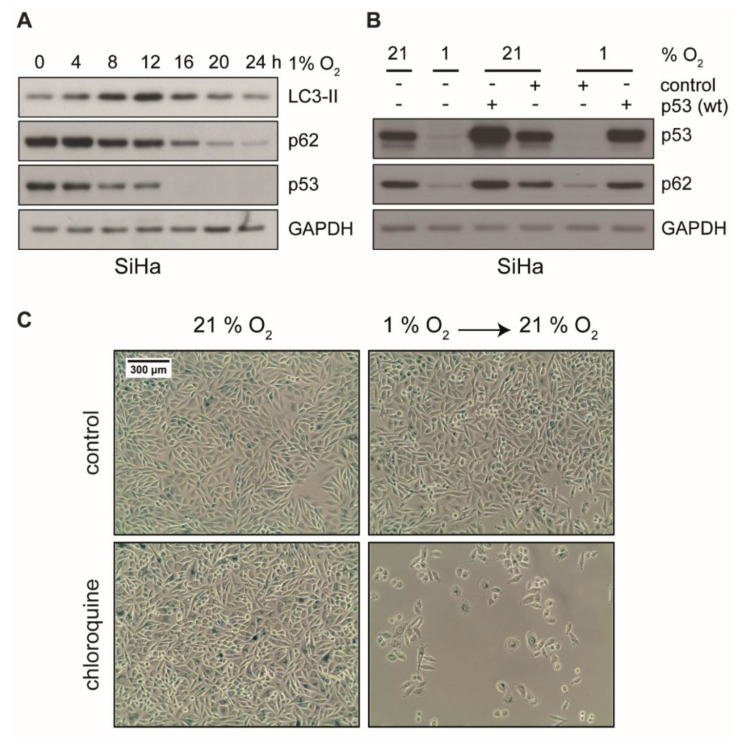
Protein levels of LC3-II, p62, and p53 in hypoxic cells and analyses of cellular senescence upon treatment with autophagy inhibitor CQ. (**A**) Western blot analyses of SiHa cells using antibodies against LC3-II, p62, and p53. After cell culturing as indicated at 1% O_2_, total protein was extracted. GAPDH served as a loading control. (**B**) Cells were transfected with an empty vector (control) or a wild type p53 expression plasmid. Subsequently, 48 h post transfection, cells were cultured under normoxia (21% O_2_) or hypoxia (1% O_2_) for 24 h prior to total protein extraction. Western blot analyses using antibodies against p53 and p62 were performed. GAPDH served as a loading control. (**C**) SiHa cells were cultured for 48 h under normoxia (21% O_2_) or hypoxia (1% O_2_), in either the absence (top) or the presence (bottom) of 50 μM chloroquine (CQ). Subsequently, cells were passaged and cultivated at 21% oxygen for additional 72 h. Cells were stained for the senescence marker SA-β-Gal. Scale bar: 300 μm.

**Figure 6 ijms-21-09533-f006:**
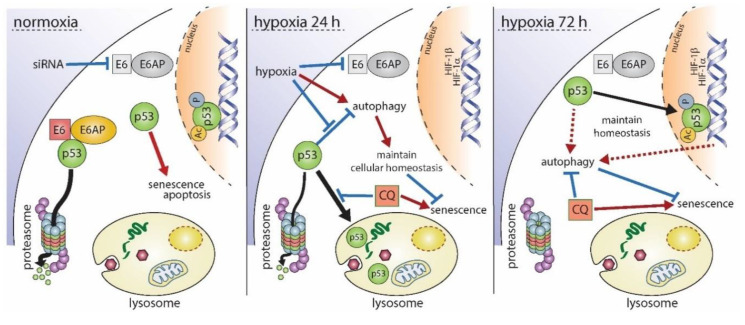
Schematic overview of the proposed biphasic regulation of p53 under hypoxia. Under normoxia (left panel), depletion of E6 causes an immediate accumulation of p53 followed by the induction of senescence or apoptosis. Hypoxic conditions for 24 h (center panel) lead to the reduction of E6 as well as p53, which is diminished by lysosomal degradation. Consequently, cells do not enter apoptosis or senescence. Autophagy, which is normally inhibited in the presence of p53, is induced under hypoxic conditions, and hypoxia-related gene expression takes place. Inhibition of p53 degradation by chloroquine (CQ) results in an irreversible growth arrest of cells. Upon prolonged hypoxia (right panel), p53 is no longer degraded, and p53-dependent as well as autophagy-related genes are expressed, allowing a maintenance of cellular homeostasis, proper autophagic flux, and inhibition of senescence, even in the presence of p53. When autophagy is inhibited by CQ under these conditions, homeostasis is disturbed resulting subsequently in the induction of senescence. Black arrows indicate a translocation of p53. Red arrows represent the induction of a cellular process whereas blue lines represent their repression, respectively.

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
