# Peer review of "p53 Is Regulated in a Biphasic Manner in Hypoxic Human Papillomavirus Type 16 (HPV16)-Positive Cervical Cancer Cells"

_ijms, 2020, doi:10.3390/ijms21249533_

Round 1
Reviewer 1 Report
In this manuscript, entitled “p53 is regulated in a biphasic manner in hypoxic human papillomavirus type 16 (HPV16)-positive cervical cancer cells “, Zhuang and colleagues, investigated the pattern of restoration of p53 protein levels during E6/E7 repression in hypoxic conditions.
A major finding of this study was that E6 depletion is associated with lysosomal degradation of p53 in high risk HPV cervical cancer cells under hypoxia. As a consequence, tumor cells escape apoptosis and senescence.
This work provides useful information about the HPV driven carcinogenesis. The manuscript is well written and the experimental work well organized.
Minor comments:
1). Authors do not discuss the efficiency of E6 silencing system that they use in normoxia results section.
2). Fig 1A, Cells have been treated with siRNA for 24h? It is not clear.
3). Is E6 silencing associated with apoptosis in hypoxia? Authors have not investigated the levels of apoptosis in their cell lines.
Author Response
Please find the detailed reply to the reviewer in the attached document.

Reviewer 2 Report
This is a well-conducted study that addresses an important question – how do HPV-positive cells survive under hypoxic conditions, particularly given the previously published observations that the viral oncogenes (E6 and E7) are downregulated in hypoxic cells. The experiments are appropriate and the results appear to be very clear. I have one minor comment regarding statistical testing (below) but I don’t expect this to change the interpretation of the data.
I’d be interested to know whether the authors have tried leaving cells in 1% O2 for more than 72 hours? It would be interesting to see whether (a) p53 levels plateau at the level to which they have recovered to by 72 hours (similar to the level seen in normoxia, in the presence of E6/E7) and (b) whether E6/E7 levels eventually recover or whether they remain low in hypoxic cells. Note, I’m not requesting that this is done as a revision but it is an interesting question to consider, I think.
The authors should correct for multiple t-testing in the analysis of the qRT-PCR data presented in Figures 1C, 3C and 4. I suggest using a one-way ANOVA with a post-hoc test (e.g. Dunnett’s correction). The results are very clear and it will not affect the assignation of statistical significance or the interpretation but would be good practice.
Author Response

(The authors gave the same response as above.)

Reviewer 3 Report
It is known that in HPV positive cervical cancer cell lines grown under normoxy the repression of E6/E7 mRNA by siRNA leads to the reconstitution of p53 and the induction of senescence. Hypoxia leads to the repression of the E6 and E7 oncogene expression. Thus an interesting question is to understand the effect hypoxia has on p53 and downstream targets in the cervical cancer cell lines. Zhuang et al. addressed this in their manuscript with the title “p53 is regulated in a biphasic manner in hypoxic human papillomavirus type 16-positive cervical cancer cells“.
The authors show here that in HPV16 positive cervical cancer cell lines SiHa and Caski grown under hypoxia p53 remains repressed at 24h followed by an increase at 72h. As consequence, the transcription of p53 responsive genes associated with senescence and apoptosis are not upregulated and the cells do not enter senescence. Yet, p53 dependent genes associated with autophagy can be induced. The results imply that hypoxia enhances cell survival and evasion of senescence by modulating p53 including its downstream network although the E6 and E7 are repressed.
The experiments are of high quality and convincing. Necessary controls are included.
The manuscript significantly contributes to clarify the role of p53 under hypoxic conditions in SiHa cells.
Yet there is one major point which should be addressed prior publication. The protein level of p53 are crucial for the message of this manuscript. Yet, there are no quantifications of the protein bands provided, neither is there any statement of how many replicates of the blots have been performed.
The authors should quantify the signals in at least two crucial blots, such as those in Fig. 1C and Fig. 2D to quantitate the level of p53 and its stabilization of MG132.
The authors should further take into account that hypoxia induces a 60% repression of p53 mRNA under these conditions, as shown in Fig. S1E.
Minor Points and comments:
- Is there any effect of hypoxia on the cell fate after incubation for 72 or even 120h?
- In Fig. 3A, the siRNA-mediated reduction of E6 /E7 mRNA is only 50%. It is astonishing that there is such a clear effect on the cell fate.
- Fig. 4A: The most right panels in the graphs Bax and Puma are labeled ns, respectively. Ns to what? This is not clear to me.
- What is the role of E6 and E7 in hypoxic cervical cancer cells? It seems that the effect is independent of the HPV oncogenes but depends on the wt p53? The authors might add one sentence in the discussion. What is known about the role of p53 on senescence and apoptosis and autophagy in other, HPV independent cancer cells grown under hypoxia?
Author Response
Please find a detailed reply to the reviewer in the attached document.

Reviewer 4 Report
The main finding of this research is the p53 reconstitution after 72 hours of hypoxia. Due to the down-regulation of E6/E7 under hypoxia, there are more p53 in hypoxia than that in normoxia. They thought that p53 might play an essential role to protect cancers by the induction of autophagy in hypoxia condition. By interruption of autophagy, the cancer cells can be introduced into senescence. This is an interesting point. However, their results did not provide enough evidences to address that. The following is my specific comments.
- They should prove that the cells really enter into autophagy after 72 hours of hypoxia. I did not see any results related to this concern in Fig 3, 4, or 5.
- I did not see the results to show that the interruption of autophagy (CQ treatment) can introduce the senescence for cancer cells. Fig 5 was too weak to claim that.
Author Response

(The authors gave the same response as above.)

Round 2
Reviewer 4 Report
No